# Involvement of Serotonergic Projections from the Dorsal Raphe to the Medial Preoptic Area in the Regulation of the Pup-Directed Paternal Response of Male Mandarin Voles

**DOI:** 10.3390/ijms241411605

**Published:** 2023-07-18

**Authors:** Zijian Lv, Lu Li, Yin Li, Lizi Zhang, Xing Guo, Caihong Huang, Wenjuan Hou, Yishan Qu, Limin Liu, Yitong Li, Zhixiong He, Fadao Tai

**Affiliations:** Institute of Brain and Behavioral Sciences, College of Life Sciences, Shaanxi Normal University, Xi’an 710062, China; lvzijian1113@163.com (Z.L.); liluli@snnu.edu.cn (L.L.); li.in@snnu.edu.cn (Y.L.); lizizhang1993@163.com (L.Z.); guoxingguoxing@snnu.edu.cn (X.G.); hch1724165936@163.com (C.H.); houwenjuan2021@snnu.edu.cn (W.H.); yishanqu@snnu.edu.cn (Y.Q.); liulimin@nwu.edu.cn (L.L.); liyitong@mail.nwpu.edu.cn (Y.L.)

**Keywords:** infanticide, paternal behaviors, MPOA, serotonin, DR

## Abstract

Male mammals display different paternal responses to pups, either attacking or killing the young offspring, or contrastingly, caring for them. The neural circuit mechanism underlying the between-individual variation in the pup-directed responsiveness of male mammals remains unclear. Monogamous mandarin voles were used to complete the present study. The male individuals were identified as paternal and infanticidal voles, according their behavioral responses to pups. It was found that the serotonin release in the medial preoptic area (MPOA), as well as the serotonergic neuron activity, significantly increased upon licking the pups, but showed no changes after attacking the pups, as revealed by the in vivo fiber photometry of the fluorescence signal from the 5-HT 1.0 sensor and the calcium imaging indicator, respectively. It was verified that the 5-HTergic neural projections to the MPOA originated mainly from the ventral part of the dorsal raphe (vDR). Furthermore, the chemogenetic inhibition of serotonergic projections from the vDR to the MPOA decreased the paternal behaviors and shortened the latency to attack the pups. In contrast, the activation of serotonergic neurons via optogenetics extended the licking duration and inhibited infanticide. Collectively, these results elucidate that the serotonergic projections from the vDR to the MPOA, a previously unrecognized pathway, regulate the paternal responses of virgin male mandarin voles to pups.

## 1. Introduction

Adult male mammals, including men, show different behavioral responses toward conspecific young offspring based on the reproductive fitness or physiological state of the individual. Child abuse and father absence are some of the leading risk factors for mental illness in human offspring [1]. Increasing paternal care and reducing male abuse may be important targets for the prevention of mental illness caused by these risk factors [2]. Mammals tend to kill or ignore unrelated neonates, but tolerate or even actively care for genetically related pups [3]. Some virgin male laboratory rodents show infanticide, while virgin females display parenting behaviors [4]. In monogamous and biparental California mice (*Peromyscus californicus*), adult males with reproductive experience consistently care for any pup to which they are exposed, while virgin males may attack, ignore, or care for other encountered pups [3,5]. In monogamous mandarin voles (*Microtus mandarinus*), some virgin males instinctively care for or lick strange pups, while other individuals kill these pups [6]. These monogamous rodents are ideal animal models for revealing the mechanism underlying the paternal response to young offspring. However, the factors contributing to individual variation in the paternal responses of virgin males remains elusive.

Individual variation in the paternal responses of males may be caused by several factors, including the genetic relationship with young offspring [7], dominance status [7], individual differences in anxiety and several neuroendocrine parameters [3], and reproductive experience [8]. Although several factors have been found to be associated with the male responses to pups, the brain mechanisms underlying between-individual variation in the pup-directed responsiveness of virgin males remain unclear.

The medial preoptic area (MPOA) has been found to regulate many types of behaviors, such as parental behaviors [9,10], sexual behaviors [11,12,13], aggression [14], preying behaviors [15], and sleep-wake patterns [16]. In addition to the involvement of MPOA in maternal behavior [17,18,19,20], the MPOA also shows higher levels of activity regarding paternal care [21,22]. MPOA lesions could also significantly increase the latency to sniff and lick pups [18]. Moreover, the optogenetic activation of the neurons in the MPOA of the virgin males restrained pup-directed aggression and increased grooming behaviors [23]. However, distinguishing the types of neurons or projections in the MPOA which regulate the paternal responses of virgin males to pups remains elusive.

One possible circuit that controls paternal responses of virgin males may be serotonin (5-hydroxytryptamine, 5-HT) projections deriving from the dorsal raphe (DR). Previous studies have found that 5-HT is involved in regulating aggressive behaviors [24,25,26]. The 5-HT1a agonist injected into the hypothalamus evokes aggressive behavior in female hamsters, but decreases aggression in their male counterparts [26]. Likewise, the over expression of 5-HT1a receptors in serotonergic neurons heightens the aggressive behaviors [24]. Some studies showed that the 5-HT1a and 5-HT1b receptors in the MPOA of the hamsters modulate their aggressive behavior [27,28,29,30,31]. In addition, pharmacological studies reveal that 5-HT1a and 5-HT2a receptors are necessary for the normal expression of maternal behavior [32,33]. However, the effects of different types of 5-HT receptors are region-specific. Whether and how 5-HT in the MPOA mediates the paternal or infanticidal behavior of virgin male voles remain unknown. Whether DR 5-HT neurons project to the MPOA and how this circuit regulates paternal behaviors also remain elusive.

The present study aimed to identify whether DR 5-HT neurons projecting to the MPOA regulate the paternal response to pups. The monogamous mandarin voles, with high levels of maternal and paternal care behaviors, were used in the present study [34,35,36,37]. Before testing, the virgin male mandarin voles were identified as paternal voles or infanticidal voles, according their behavioral response when exposed to alien pups. These between-individual variations in pup-directed paternal responses provided an opportunity to reveal the neural mechanisms underlying paternal care [38].Then, the real-time release of 5-HT in the MPOA of males upon encountering pups was detected using 5-HT 1.0 sensor virus, which can emit a fluorescent signal upon binding with 5-TH, and the fluorescent signals were subsequently examined using in vivo fiber photometry [39]. The 5-HT neuron projections from the DR to the MPOA were determined using retrograde tracer cholera toxin subunit B (CTB). The real-time activity of the 5-HTergic neurons in the DR projecting to the MPOA was displayed by the intensity of the fluorescent signals of the calcium indicator GCaMP6m and was measured by in vivo fiber photometry. Next, chemogenetic and optogenetic methods were used to inhibit or activate the 5-HTergic neurons in the DR projecting to the MPOA, and their effects on paternal or infanticidal behaviors were examined. This study elucidates a previously unrecognized pathway, serotonergic projections from vDR to MPOA, that may mediate the paternal response of virgin males to alien pups. In addition, this study may provide a new target for the reduction of child maltreatment or abuse and an increase in paternal engagement for the therapeutic intervention of mental illness caused by these abnormal parental responses.

## 2. Results

### 2.1. 5-HT Release in the MPOA upon Licking or Attacking

We utilized the 5-HT 1.0 sensor to monitor the release of 5-HT in the MPOA during different behaviors of male voles (Figure 1A–C). Meanwhile, the control virus, which only encode fluorescence protein, was used to validate the 5-HT 1.0 sensor (Appendix A). According to the results of delta F/F for different behaviors, the levels of 5-HT release in the MPOA did not change, before or after retrieval (Figure 1D_1_) and sniffing (Figure 1E_1_), in paternal voles and infanticidal voles. Meanwhile, the heat maps showed no changes in colors among the behaviors. The quantitative comparison of the mean of the delta F/F for the pre and post retrieval and sniffing behaviors did not reveal any difference between the paternal and infanticidal voles (Figure 1D_2_,D_3_,E_2_,E_3_). In addition, the areas under the curve (AUC) for the retrieval and sniffing behaviors showed no difference between the paternal and infanticidal voles (Figure 1D_4_,E_4_).

However, the delta F/F of the paternal voles increased markedly after licking the pups, and this increase was maintained for at least 2 s (Figure 1F_1_). The mean values of delta F/F increased significantly upon licking the pups, indicating a significant increase in 5-HT (Figure 1F_2_). Additionally, the curve of the delta F/F and the heat map did not change when attacking the pups, and the mean value of the delta F/F indicated that the release of 5-HT did not change upon attacking the pups (Figure 1F_1_,F_3_). Moreover, depending on the result of the AUC, the levels of 5-HT secretion in the MPOA were significantly different between the paternal and infanticidal voles.

Furthermore, the fluorescence signal of the control virus of the 5-HT 1.0 sensor did not change during retrieving, sniffing, licking, and infanticidal behaviors, and the 5-HT 1.0 sensor was validated for monitoring the 5-HT release in the MPOA (Appendix A).

### 2.2. Verification of 5-HT Neuron Projections from the Ventral Part of the DR Region to the MPOA

To determine whether 5-HT neurons from the DR or median raphe (MR) project into the MPOA in mandarin voles, retrograde tracers (CTB) were injected into the MPOA (Figure 2A). As a retro neural circuit indicator, CTB could transport backwards, upstream of the projections. After CTB was allowed to diffuse, the MPOA and DR sections were collected and stained with an immunofluorescent (Figure 2B).

Depending on the number of TPH2 and CTB merged neurons in the DR, we could conclude that the 5-HT neuron projections into the MPOA originated mainly from the ventral area of the DR (Figure 2C–E).

### 2.3. Activities of 5-HTergic MPOA Projecting Neurons in the DR Increase upon the Licking of Pups

To investigate whether the 5-HTergic neurons in the DR projecting to the MPOA would be activated when licking or attacking the pups, we injected rAAV-Ef1α-DIO-GCaMP6m, a genetically encoded fluorescent Ca^2+^ sensor, into the DR region and Retro/rAAV-TPH2-CRE into the MPOA. After three weeks, the optical fiber was implanted into the DR region to collect the fluorescent signals (Figure 3A). The combination of DIO-GCaMP6m and TPH2-CRE allows for GCaMP6m protein to be expressed on the 5-HTergic neurons in the DR projecting to the MPOA (Figure 3B). Post hoc immunofluorescent analysis revealed that 63.48% Gcamp6^+^ cells were positive for TPH2^+^ in the ventral part of the DR. The control group underwent the same process as the GCaMP6m group (Appendix A).

We found that the 5-HTergic neurons displayed higher levels of activity during the licking of pups and remained at a high activity level at the onset and for the duration of the licking behavior (Figure 3E_1_,E_2_). However, the GCaMP fluorescence signal of the 5-HTergic neurons decreased during pup retrieving (Figure 3C_1_). The mean of delta F/F also indicates that the 5-HTergic neurons were inactive during the retrieval behavior (Figure 3C_2_). On the contrary, the GCaMP fluorescence signal of the 5-HTergic neurons in the DR region did not change upon sniffing. The curve of the delta F/F and the color of the heat map did not change during the sniffing behavior (Figure 3D_1_). The comparison of the mean of delta F/F, before and after sniffing, confirmed the above results (Figure 3D_2_). Meanwhile, the GCaMP fluorescence signal of the 5-HTergic neurons did not change during infanticidal behavior (Figure 3F).

Apart from that, the activities of the neuron in the control virus group did not show significant differences before or after the occurrence of each behavior (Appendix A). Fluorescent changes induced by artifacts can be ruled out.

### 2.4. Chemogenetic Inhibition of the 5-HTergic Neural Projections Reduces the Licking Duration and the Latency to Attack Pups

In order to inhibit 5-HTergic neural projections from the DR to the MPOA, rAAV-Ef1α-DIO-hM4Di and Retro/rAAV-TPH2-CRE were injected into the DR and MPOA regions, respectively (Figure 4A and Appendix A). Then, the behaviors of paternal and infanticidal voles were recorded after the injection of saline or CNO (Figure 4B). The results of in vitro whole-cell patch clamp recordings validated the function of the hM4Di virus in the 5-HTergic neural projections (Figure 4C). Post hoc immunofluorescent staining indicated that 67.07% of the hM4Di positive cells overlapping with the TPH2 positive cells originated from the ventral portion of the DR (Figure 4D,E).

When the paternal voles received CNO intraperitoneally, the licking duration of paternal voles decreased significantly (Figure 4F). Furthermore, CNO injection increased the neglect of pups in hM4Di-expressing paternal voles. However, the duration of pup retrieval pups was decreased (Appendix A). The paternal group receiving the control virus did not show any differences in regards to the behaviors of paternal voles, before and after injecting CNO or saline. Moreover, the latency to attack pups was shortened after the injection of CNO in infanticidal voles receiving the hM4Di virus (Figure 4G). Injection of CNO or saline did not produce any effect on the behaviors of infanticidal voles receiving the control virus.

### 2.5. Optogenetic Activation of 5-HTergic Neural Projections from DR to MPOA Increases the Licking Duration and Elongates the Latency to Attack Pups

To activate 5-HTergic neural projections from the DR to the MPOA using the optogenetic method, rAAV-Ef1α-DIO-ChR2 and Retro/rAAV-TPH2-CRE were injected into the DR and MPOA, respectively. The control animals were injected with the mCherry virus, which did not encode ChR2 protein (Figure 5A). Before the behavior testing, the optical fibers were implanted into the MPOA to activate the terminal of the 5-HTergic neural projection (Figure 5B). The results of in vitro whole-cell patch clamp recordings validated the function of the ChR2 virus (Figure 5C). Post hoc immunofluorescent staining revealed that 61.11% of ChR2 positive cells were also positive for TPH2 and were mainly distributed in the ventral part of the DR region (Figure 5D,E and Appendix A).

The results showed that the light stimulation of DR 5-HTergic neurons projecting into MPOA increased the licking duration of paternal voles, while this manipulation produced no effect on the control group (Figure 5F). Meanwhile, the latency to attack the pups was elongated after the light stimulation. The infanticidal behavior would reappear while the light was off (Figure 5G). Furthermore, the infanticidal voles tended to neglect the pups and leave them alone when the 5-HTergic neural projections were activated (Appendix A). However, this manipulation did not produce any effect on the control group.

## 3. Discussion

In this study, we identified an important function of 5-HTergic neural projections from the DR to the MPOA in regulating the pup-directed paternal responses in virgin males. Using the 5-HT 1.0 sensor, we found that 5-HT release significantly increased upon licking the pups. Then, we verified that 5-HT neurons in the DR, especially the ventral area, projected to the MPOA. The 5-HTergic neurons in the DR projecting to the MPOA displayed a higher level of activity upon the licking of the pups, which was maintained for a long period of time. Chemogenetic inhibition of the serotonergic projections from the DR to the MPOA of the virgin males reduced pup licking and the latency to attack pups, while activating these projections produced the opposite effects. Our study puts forward a new potential circuit mechanism in which 5-HT neurons in the DR projecting to the MPOA could regulate the pup-directed paternal responses of virgin males.

In the present study, the inhibition of these projections in the MPOA of virgin males reduced licking and the latency to attack pups, and the activation of these projections produced the opposite effects. In previous study, c-Fos expression in the MPOA increased upon paternal care in male mice [40,41]. Likewise, the electrolytic lesion of MPOA of ICR mice could extend the latency of pup retrieval, grooming, and crouching behaviors [42]. Additionally, the optogenetic activation of GABAergic neural projections from the MPOA to the BNSTrh could attenuate the infanticide of virgin male mice [40]. The lesion of the MPOA region could reverse the mating-induced transformation of the infanticidal to paternal behaviors [43]. Although these studies reveal an important role of the MPOA in mediating paternal and infanticidal behaviors, which were supported by our results, the present study identifies a new unrecognized pathway from the DR to the MPOA that regulates the pup-directed paternal responses of virgin males.

Another finding is that the 5-HTergic neurons projecting to the MPOA mainly originated from the ventral part of the DR regions. This result is consistent with the study in which Fluor-Gold was used as a fluorescent retrograde tracer, and projections from the DR to the MPOA were found in female Wistar rats [44]. Another study found that neurons in the median raphe nucleus of ewes project to the MPOA using the retrograde fluorogold as tracer [45]. In addition, the results of the electrolytic lesion of the median or dorsal raphe nucleus in male Sprague Dawley rats showed that the median raphe nucleus was the main source of 5-HT fibers to the anterior hypothalamic area and the MPOA [46]. These findings show some differences with those of the present study, which revealed that the 5-HT fibers to the MPOA were derived from the ventral part of the DR. One possible reason may be the difference in the sex and species studies, as male mandarin voles were used in the present study, whereas female sheep and rats were used in previous studies.

Accumulating studies have investigated the involvement of 5-HT in parenting behaviors. It has been found that 5-HT alters maternal behaviors and increases the affiliative behaviors of males [32,47]. The 5-HT neurons projecting from the DR to the anterior hypothalamus play a vital role in the maintenance of the maternal behaviors of postpartum rats [48]. In addition, the 5-HTergic neurons in the caudal DR of male California mice were active upon mice exposure to pups [49]. Although an association between 5-HT and maternal care has been revealed, the present study revealed a new finding that the activation of 5-HTergic neural projections from the DR to the MPOA could increase the licking behavior of male mandarin voles.

Our study shows that the chemogenetic inhibition of the 5-HTergic neural projection from the DR to the MPOA reduced the latency to attack pups. In contrast, the optogenetic activation of 5-HT neural projection from the DR to the MPOA increased the latency to attack the pups. This result is consistent with those from a previous study which showed that the hypothalamic injection of the 5-HT1a agonist could inhibit aggression in male Syrian hamsters [26]. Meanwhile, the suppression of the serotonin neuron firing increased the aggression level of male mice [24]. Both the 5-HT1a and 5-HT1b agonists injected into the MPOA could decrease the aggressive behaviors of CF-1 male mice [50]. These previous reports support our finding that 5-HTergic neuron projections from the DR to the MPOA regulate infanticidal behaviors

In the present study, the activation of 5-HTergic neurons in the DR projecting to the MPOA reduced the motivation for infanticide, but did not transform infanticidal to paternal behaviors. This may be because that MPOA has different upstream regions that control different behaviors. For example, MeA, an upstream region of MPOA, also regulates the pup-directed behaviors of male mice. The virgin male mice tend to attack, but not to groom, pups during the high activation of GABAergic neurons in the MeA, while this effect was not seen in female mice [51,52]. Thus, to completely alter pup-directed behavior from infanticidal to paternal in virgin male voles, additional brain regions need to be manipulated.

In summary, this study identifies the 5-HTergic neural projections from the DR to the MPOA as the previously unrecognized circuit in control of the pup-directed behavioral responses of virgin male mandarin voles. Although paternal behaviors toward their own pups were observed directly to investigate paternal care in some studies, most studies examining paternal behavior used virgin male subjects [8]. We determined that the activation of the 5-HTergic neural projections from the DR to the MPOA of virgin male voles could increase pup licking and attenuate pup attacking, while the inhibition of these projections produced the opposite effects, suggesting a possible mechanism underlying the transition from virgin male to father. This finding provides a new circuit mechanism underlying the paternal and infanticidal behavior of virgin males. In the future, the manipulation of this pathway using pharmacological or other methods could possibly be used to reduce child abuse and increase paternal engagement to prevent or intervene in the mental illness of offspring caused by abnormality in paternal responses. Thus, the present finding also provides a new target for the prevention or treatment of some psychiatric illnesses associated with abnormal parental responses to offspring, such as child maltreatment or abuse, as well as decreased levels of paternal care.

## 4. Material and Methods

### 4.1. Animals

Mandarin voles used in the study were laboratory-housed offspring collected from a wild population originating from Henan, China. Animals were housed under a 12:12 light: dark cycle, with ample food (carrots, grain feed) and water, with three voles per one clean polycarbonate cage (44 cm × 22 cm × 16 cm). Each procedure used complied with the guidelines of the Animal Care and Use Committee of Shaanxi Normal University and followed the Guide for the Care and Use of Laboratory Animals of China.

The male mandarin voles were removed from their parents at postnatal day 24 and were housed with same-sex litter mates in the new cages for about two months. Before the behavioral tests, the virgin male voles were given thirty minutes to acclimatize to the test room, with an 80 lux illumination. Then, an unrelated 7-day-old pup was then placed into the cage with a virgin male mandarin vole, and all behavioral responses of the male voles were recorded for 10 min. The males that licked, groomed, or crouched over the pups were considered as paternal voles, while the males that attacked or bit the pups were regarded as infanticidal voles. The paternal voles showed behaviors involving approaching, sniffing, retrieving, licking, grooming, and crouching. The infanticidal voles showed behaviors including approaching, sniffing, retrieving, and attacking pups. All behaviors in different tests were recorded by video recorder between 9 a.m. and 2 p.m. The data were scored and analyzed by a single observer, blinded to the experimental condition, using Jwatcher V1.0 software (http://www.jwatcher.ucla.edu; accessed on 1 May 2023). The detailed scheme of the entire experiment is detailed in the Appendix A).

### 4.2. Stereotaxic Surgery

Adult virgin mandarin voles were anesthetized using isoflurane and were fixed with a stereotaxic apparatus (R.W.D. Life Science, Shenzhen, China). Next, a Hamilton syringe was used to deliver the virus into the brain regions. The injection velocity was controlled at 100 nL/min. Before it was slowly withdrawn, the needle was left in 0.1 mm above the target coordinate for 5 min in order to allow for the diffusion of the virus. The bregma was used as a reference point for locating the brain region.

To examine the real-time release of 5-HT in the MPOA, the 5-HT 1.0 sensor virus (rAAV-hSyn-5HT2.1-WPRE-Hgh, 300 nL, Titer: 2.53 × 10^12^ vg/mL, BrainVTA, Wuhan, China) was injected into the MPOA (AP 0.1 mm, ML + 0.3 mm, DV—5.3 mm), and the optical fibers were implanted into the MPOA (AP 0.1 mm, ML + 0.3 mm, DV—5.1 mm). To determine the origin of the 5-HT neuron projections, Cholera Toxin Subunit B (CTB) (555, 200 nL, Thermo Fisher Scientific, Waltham, MA, USA) was also injected into the MPOA. To monitor the real-time activity of the 5-HT neurons in the DR, the calcium imaging virus (rAAV-EF1α-DIO-Gcamp6M-WPRE-pA, 600 nL, Titer: 5.15 × 10^12^ vg/mL, BrainVTA) and the retro TPH2 virus (rAAV(Retro)-TPH2-Cre-WPRE-pA, 300 nL, Titer: 5.83 × 10^12^ vg/mL, BrainVTA) were injected into the DR (AP—4.36 mm, ML 0 mm, DV—3.3 mm) and the MPOA (AP 0.1 mm, ML + 0.3 mm, DV—5.3 mm), respectively. The retro TPH2 virus can retrogradely transfect the DR neurons. The optical fibers were implanted in the DR (AP—4.36 mm, ML 0 mm, DV—3.1 mm). To manipulate the 5-HT neuron projecting from the DR to the MPOA, the chemogenetic virus (rAAV-Ef1α-DIO-hM4D(Gi)-mCherry-WPREs, 600 nL, Titer: 2.09 × 10^12^ vg/mL, BrainVTA) and the optogenetic virus (rAAV-Ef1α-DIO-ChR2-mCherry-WPRE-pA, 600 nL, Titer: 2.01 × 10^12^ vg/mL, BrainVTA) were injected into the DR, while the retro virus was injected into the MPOA. Additionally, the control virus for each experiment was also injected into the corresponding site in the control subjects (rAAV-hSyn-EGFP-WPRE-hGH-pA for the 5-HT sensor experiment, 300 nL, Titer: 2.00 × 10^12^ vg/mL; rAAV-Ef1α-DIO-mCherry-WPRE-pA for the chemogenetic and optogenetic experiments, 600 nL, Titer: 2.02 × 10^12^ vg/mL, BrainVTA). After the surgery, all male voles were returned to their home cages for recovery for at least one week.

### 4.3. Fiber Photometry

To detect the fluorescence signal from the 5-HT 1.0 sensor and the calcium imaging indicator, an optical fiber (250 μm O.D., NA = 0.37; length 6.5 mm for MPOA, 4.5 mm for DR) was implanted into the MPOA or DR one week before the fluorescence emission was recorded. The voles were housed in their own cages for recovery.

To record the fluorescence signals, a cluster of beams, launched from a 488 nm laser (OBIS 488LS; Coherent, Santa Clara, CA, USA), was reflected by a dichroic mirror (MD498; Thorlabs, Newton, NJ, USA), focused by a 10x objective lens (NA = 0.3; Olympus, Tokyo, Japan). An optical fiber (200 μm O.D., NA = 0.37, length of 1 m) guided the light to the implanted optical fiber. On the test day, the 5-HT 1.0 sensor and calcium signal fluorescence emission was band pass filtered (MF525-39, Thorlabs) and collected by a photomultiplier tube (R3896, Hamamatsu, Shizuoka, Japan). An amplifier (C7319, Hamamatsu) was used to convert the photomultiplier tube current output into voltage signals, which were further filtered through a low-pass filter (35 Hz cut-off; Brownlee 440). The analog voltage signals were digitalized at 50 Hz and recorded by fiber photometry software (Thinkerbiotech, Nanjing, China). Then, the resulting signals were analyzed using custom-written MATLAB software (Thinkerbiotech, Nanjing, China), while the area under the curve (AUC) per second of ΔF/F and the mean of ΔF/F were calculated using the SPSS software; pictures were drawn using GraphPad Prism 8 software.

### 4.4. Immunofluorescence

The immunofluorescence method was used to reveal the virus transfection, the CTB distribution, the TPH2 positive neurons and the location of the fiber and syringe needle. The experimental mandarin voles were anesthetized using 2% pentobarbital sodium (0.02 mg/mL). The brains were removed and stored in 4% paraformaldehyde for 7 d at 4 °C. The brains were then dehydrated twice in 30% sucrose solution at 4 °C. The brains were sliced by a freeze microtome (CM 1950, Leica, Wetzlar, Germany) and then mounted on slides, with a slice thickness of 0.04 mm. All vole brain coronal slices were collected and stored at −20 °C.

The sections on slides were dried at room temperature, washed with 0.1 M phosphate buffer solution (PBS) for 10 min, incubated in hydrogen peroxide for 20 min, and washed for 3 × 5 min with PBS. Then, the sections were incubated in 0.1% Triton-X 100 for 20 min and directly blocked with 5% BSA blocking solution for 60 min at room temperature. Next, the sections were incubated with the primary antibody goat Anti-TPH2 (1:500, Abcam, Cambridge, UK, ab121013) for at least 48 h at 4 °C, washed for 3 × 5 min with PBS, and then incubated with a second round of antibodies, Alexa Fluor 488 Donkey Anti-Goat IgG 1:200, Jackson, Noida, India, 705-545-147,) for at least 1.5 h in the dark. Then, the nuclei were stained with DAPI dye for 15 min. The sections were then washed for 3 × 5 min with PBS and fixed with antifade solution. Finally, the sections were observed through a fluorescent microscope (FV-1000, Olympus, Tokyo, Japan).

### 4.5. Chemogenetics

For chemogenetic inhibition, the Gi virus was injected into the DR, while the retro virus was injected into the MPOA. In addition, a control group was injected with the control virus and the retro virus. All injections were conducted with a delivering velocity of 100 nL per min using a microsyringe (10 μL, Hamilton, Bonaduz, Switzerland). Then, the needle was retained 0.1 mm above the injection site for 5 min to allow for virus diffusion. Finally, the needle was removed, and the wound was closed with nylon sutures.

Before the male voles were exposed to pups in their own cages, they were intraperitoneally injected with clozapine-N-oxide (CNO, Cat# CNO-02, 1 mg/kg, BrainVTA) prior to behavioral testing. CNO, which can inactivate the human M4 muscarinic DREADD receptor coupled to Gi, was dissolved in 99.5% dimethyl sulfoxide (DMSO, Sigma-Aldrich, St. Louis, MO, USA) and diluted with 0.9% saline to a final concentration of 1 mg/mL. All behavioral tests took place 30 min after the injection of CNO or saline, and the responses of the virgin male voles were scored and analyzed, as described above.

### 4.6. Optogenetics

All adult male mandarin voles were injected with the ChR2 virus at 10 weeks of age and were allowed to stay in their own home cages for at least five weeks for recovery and virus transfection. The fibers were implanted 0.1 mm above the virus injection site 2 weeks after virus injection. For optogenetic activation, the ferrules were linked to a 473 nm laser diode through an FC/PC adaptor and a fiber optic rotary joint. The output parameters were 10 ms, 30 Hz, with an 8 s on and 2 s off cycle, using ~10 mW for terminal stimulation.

For behavioral testing, the voles were exposed to the pups three times (5 min each), with an inter-exposure interval of 5 min. The laser light was off during the first and third encounters, but it was turned on during the second exposure. The behaviors of the voles were scored and analyzed by a single observer, who was blinded to the experimental condition, using Jwatcher V1.0 software.

### 4.7. In Vitro Electrophysiology Recordings

To verify the effects of the chemogenetic inhibitions and the optogenetic activation, in vitro whole-cell patch-clamp recordings were used to determine the activity of the DR neurons. Neurons expressing hM4Di were visually recognized by the mCherry fluorescence signal protein. The mandarin voles were anesthetized by isoflurane. The brains were dissected and cut into 300 μm coronal slices containing the DR or the MPOA. All slices were prepared in a chamber filled with artificial cerebrospinal fluid (ACSF) (32–34 °C) using vibratome (Campden, Gloucestershire, UK, 7000 smz). The data were obtained using a multiclamp 700B amplifer, filtered at 5 kHz, and sampled at 10 kHz with a Digidata 144A. Clampex 10.5 was used for the analysis.

The evoked action potentials were measured by current-clamp recordings. For CNO inhibition, the spontaneous firing of action potentials in the DR neurons at −60 mV was recorded using the current-clamp mode. After 5 min of baseline recording, the slices were perfused with 10 μm CNO. The total recording time for each cell was 10 min. For light activation, the light protocols used during behavioral testing were delivered through a 200 mm optical fiber close to the recorded neurons.

### 4.8. Statistical Analysis

All data were analyzed using SPSS version 26 (SPSS Institute, Chicago, IL, USA), and one-sample Kolmogorov–Smirnov tests were used to assess normality. The results are shown as mean ± SEM. The data from the 5-HT sensor and the calcium imaging test were analyzed using a paired sample *t*-test. In addition, the results from the chemogenetic and optogenetic testing were analyzed using a paired sample *t*-test. Additionally, the AUC was calculated using GraphPad prism 8 and analyzed by an independent sample *t*-test. The significance level was set at *p* < 0.05, and more detailed analysis methods are showed in the legend of each figure.

## Figures and Tables

**Figure 1 ijms-24-11605-f001:**
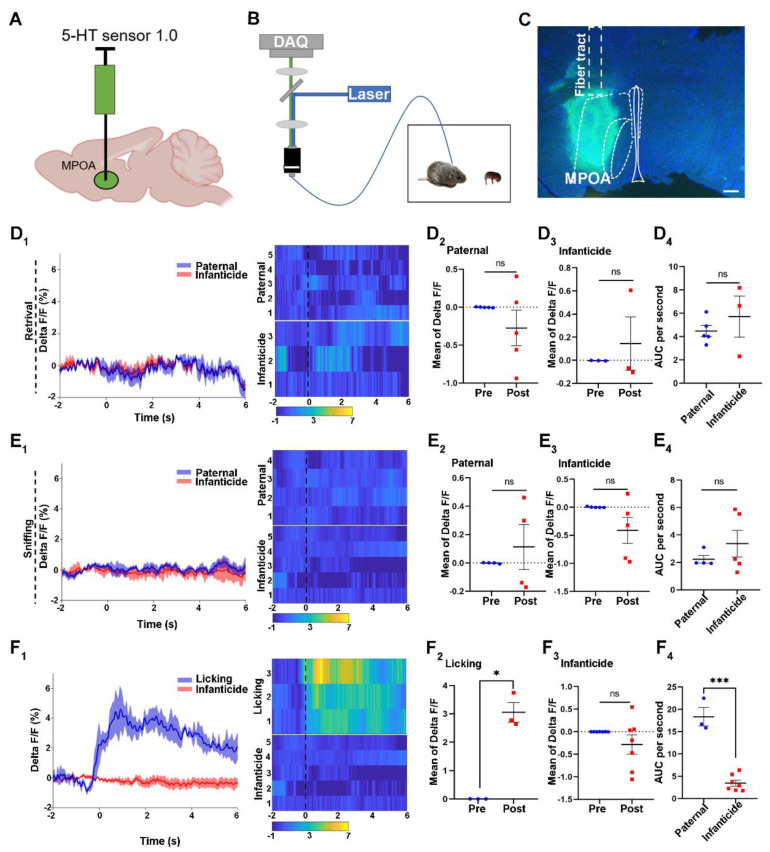
The 5-HT release upon licking or infanticidal behavior of virgin adult male mandarin voles. (**A**,**B**) The fiber photometry system used for recording the 5-HT signal from the MPOA. Schematic of behaviors assay (**B**); injection strategy (**A**); (**C**) representative injection site and fiber implantation for fiber photometry recording. Scale bar, 200 μm; (**D_1_**,**E_1_**,**F_1_**) Average 5-HT signal changes during retrieving: (**D_1_**) sniffing, (**E_1_**) infanticide, and (**F_1_**) licking. Meanwhile, the heat maps also represented each behavior. (**D_1_**) voles showing paternal care = 5, voles showing infanticide = 3; (**E_1_**) voles showing paternal care = 4, voles showing infanticide = 5; (**F_1_**) voles showing licking = 3, voles showing infanticide = 5. (**D_2_**,**D_3_**,**E_2_**,**E_3_**,**F_2_**,**F_3_**) Quantification of change in 5-HT sensor fluorescence signals before and after pup retrieval (**D_2_**,**D_3_**); sniffing (**E_2_**,**E_3_**); licking (**F_2_**); infanticide (**F_3_**). Paired sample *t*-test. (**D_2_**) *n* = 5 voles, t (4) = 1.16, *p* = 0.311 > 0.05; (**D_3_**) *n* = 3 voles, t (2) = −0.638, *p* = 0.588 > 0.05; (**E_2_**) *n* = 4 voles, t (3) = −0.721, *p* = 0.523 > 0.05; (**E_3_**) *n* = 5 voles, t (4) = 1.775, *p* = 0.151 > 0.05; (**F_2_**), *n* = 3 voles, t (2) = −8.732, *p* = 0.013 < 0.05; (**F_3_**) *n* = 7 voles, t (6) = 1.324, *p* = 0.234 > 0.05. (**D_4_**,**E_4_**,**F_4_**) Comparison of area under the curve (AUC) per second of retrieval: (**D_4_**) sniffing; (**E_4_**) licking; and (**F_4_**) infanticide. Independent sample *t*-test: (**D_4_**) t (6) = 0.355, *p* = 0.735 > 0.05; (**E_4_**) t (7) = −1.004, *p* = 0.349 > 0.05; (**F_4_**) t (8) = 9.043, *p* < 0.001. Changes in the fluorescence signal of the control virus of the 5-HT 1.0 sensor during different behaviors are shown in Appendix A. The data are shown with the mean ± SEM. ns, not significant; * *p* < 0.05; *** *p* < 0.001.

**Figure 2 ijms-24-11605-f002:**
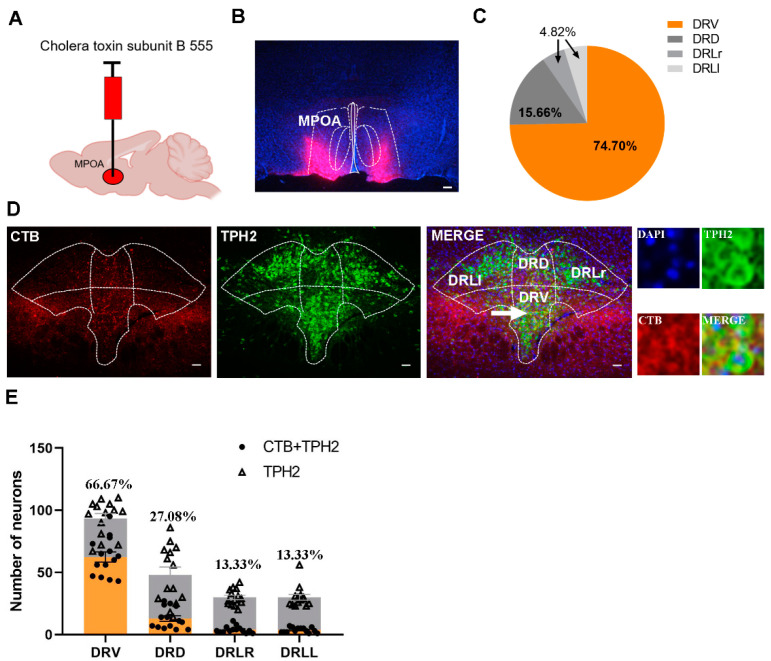
Verification of 5-HT neuron projections from the ventral part of the DR region to the MPOA. (**A**) Injection strategy of CTB; (**B**) the representative distribution of CTB in MPOA; (**C**) the percentage of CTB + TPH2 cell numbers in different subregions of the DR in the total number of CTB + TPH2 cells in the DR. DRD: dorsal part of DR; DRV: ventral part of DR; DRLr: right lateral part of DR; DRLl: left lateral part of DR. (**D**) The representative images of the merge of 5-HT neurons and CTB in the DR; (**E**) the number of CTB + TPH2 cells and TPH2-positive cells in each subregion of DR and the percentages of CTB + TPH2 cells in TPH2 cells of each subregion. The data are shown as mean ± SEM. Scale bar, 200 μm.

**Figure 3 ijms-24-11605-f003:**
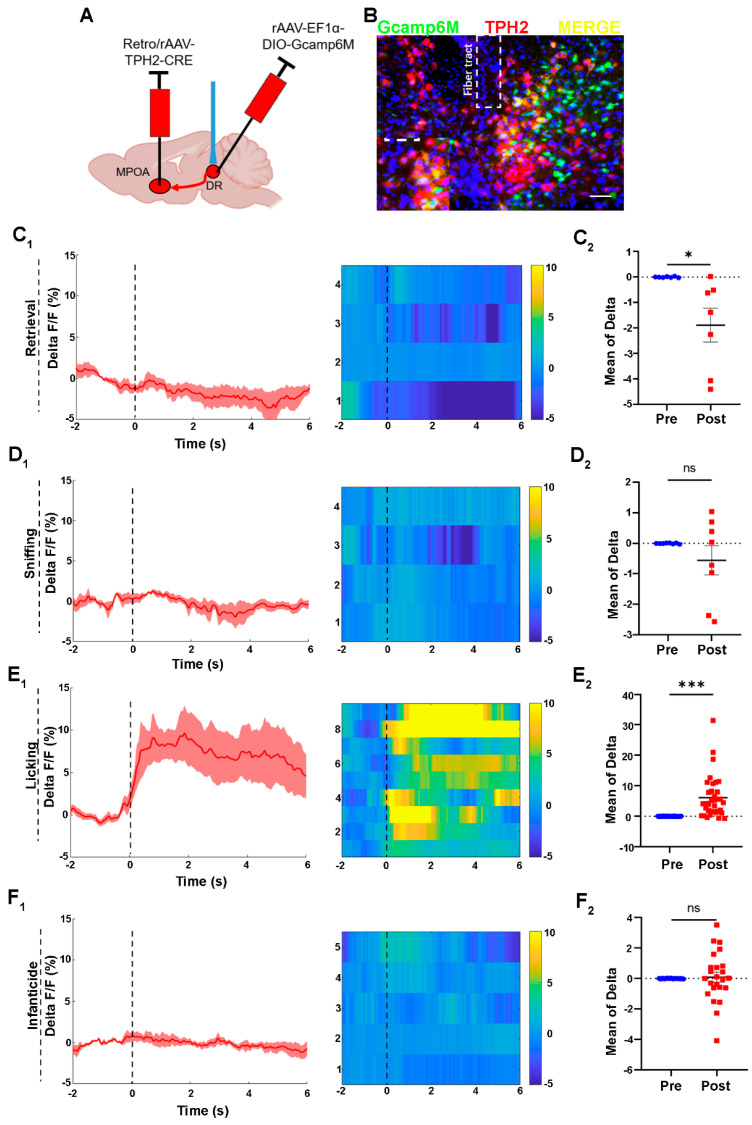
Activities of 5-HTergic neurons in the DR during the licking behavior of voles. (**A**) The injection strategy for the calcium imaging virus; (**B**) the representative photomicrograph of calcium virus expression in the DR; Scale bar, 200 μm. (**C**) The activities of neuron during retrieval behaviors of paternal and infanticidal voles; (**C_1_**) the curve of delta F/F, heat map; *n* = 4 voles. (**C_2_**) The mean of delta F/F, paired sample *t*-test, *n* = 7, t (6) = 2.829, *p* = 0.030 < 0.05; (**D**) the activities of neurons during sniffing behaviors of paternal and infanticidal voles; (**D_1_**) the curve of delta F/F, heat map; *n* = 4 voles; (**D_2_**) the mean of delta F/F, paired sample *t*-test, *n* = 8, t (7) = 1.156, *p* = 0.286 > 0.05; (**E**) the activities of neurons during licking behaviors of paternal voles; (**E_1_**) the curve of delta F/F, heat map; *n* = 9 voles; (**E_2_**) the mean of delta F/F, paired sample *t*-test, *n* = 35, t (34) = −5.263, *p* < 0.001; (**F**) the activities of neurons during infanticidal behaviors of infanticidal voles; (**F_1_**) the curve of delta F/F, heat map. *n* = 5 voles; (**F_2_**) the mean of delta F/F, paired sample *t*-test, *n* = 22, t (21) = −0.007, *p* = 0.995 > 0.05. Changes in the fluorescence signal of the control virus of GCaMP6m during different behaviors are displayed in Appendix A. The data are showed as mean ± SEM; ns, not significant; * *p* < 0.05; *** *p* < 0.001.

**Figure 4 ijms-24-11605-f004:**
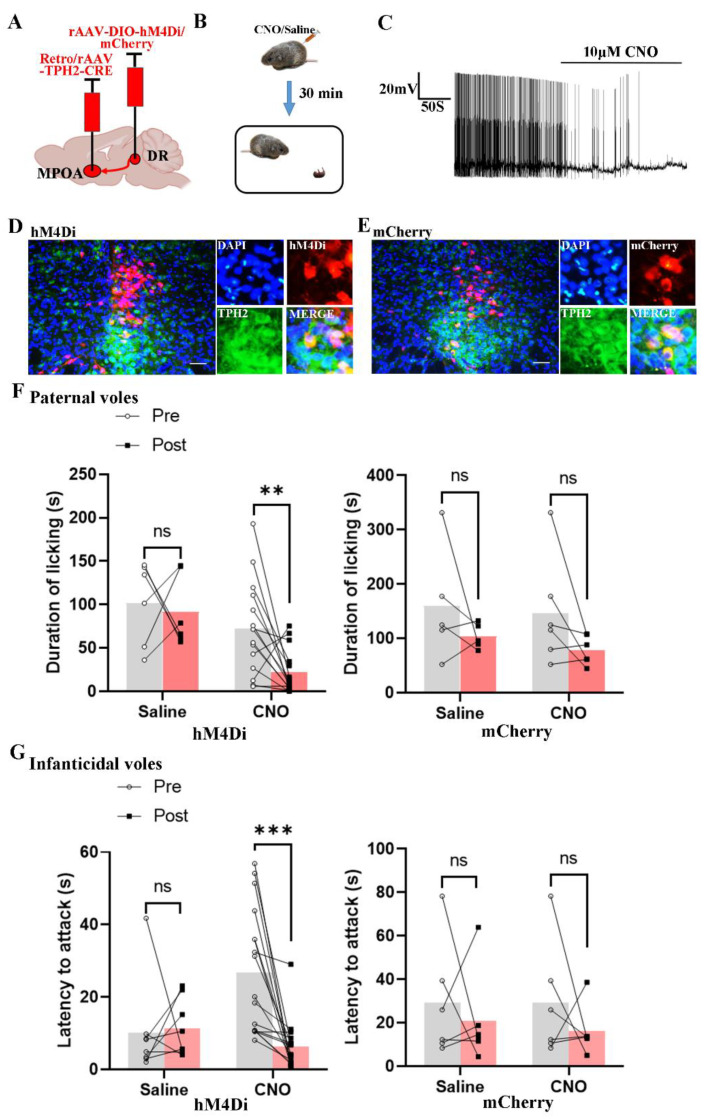
Chemogenetic inhibition of the 5-HT neural projections decreased the licking duration and shortened the onset latency of infanticide of voles. (**A**) The injection strategy of the hM4Di and mCherry virus; (**B**) the protocol of behaviors test; (**C**) the representative trace from a Gi-DREADD neuron; (**D**,**E**) the expression of the hM4Di and mCherry virus in the DR region; (**F**) the results of hM4Di-expressing and mCherry-expressing paternal voles, before and after injecting CNO or saline. Paired sample *t*-test. hM4Di voles, t saline (5) = 0.31, *p* = 0.769 > 0.05; t CNO (14) = 3.14, *p* = 0.007 < 0.01. mCherry voles, t saline (4) = 1.128, *p* = 0.322 > 0.05; t CNO (5) = 1.842, *p* = 0.125 > 0.05. (**G**) The results of hM4Di-expressing and mCherry-expressing infanticidal voles, before and after injecting CNO or saline. Paired sample *t*-test. hM4Di voles, t saline (7) = −0.177, *p* = 0.864 > 0.05; t CNO (16) = 4.394, *p* < 0.001. mCherry voles, t saline (5) = 0.548, *p* = 0.607 > 0.05; t CNO (5) = 0.951, *p* = 0.385 > 0.05. The effects of chemogenetic inhibition of DR to MPOA 5-HTergic projection on other behaviors are shown in Appendix A. The are data shown as mean ± SEM. Scale bar, 200 μm; ns, not significant; **, *p* < 0.01; ***, *p* < 0.001.

**Figure 5 ijms-24-11605-f005:**
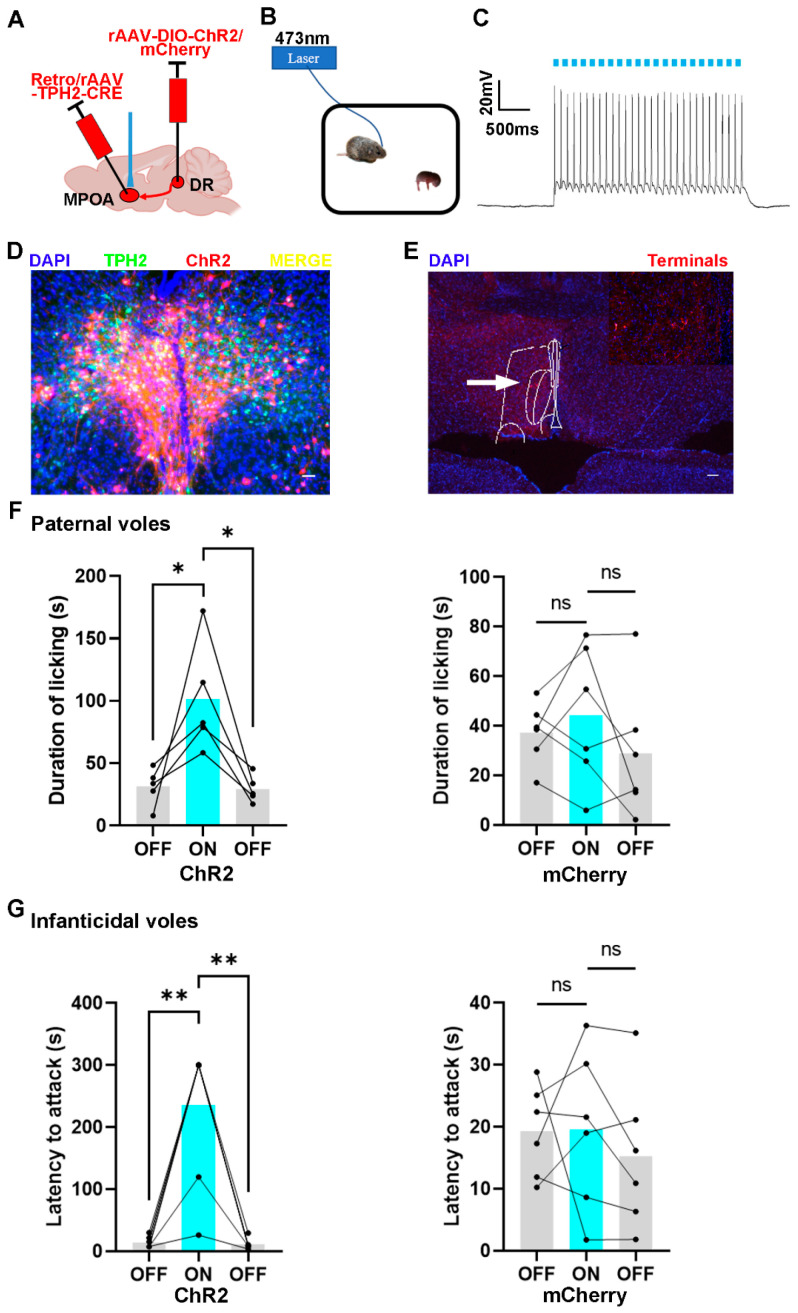
The effects of optogenetic activation of DR to MPOA 5-HTergic projections on the licking duration and the onset latency of infanticide of voles. (**A**) The injection strategy of the ChR2 and mCherry virus; (**B**) the protocol of the behavior test; (**C**) the representative trace from a ChR2-expressing neuron; (**D**,**E**) the expression of the ChR2 virus in the MPOA and DR regions. (**D**) scale bar 200 μm; (**E**) scale bar 400 μm; (**F**) the results of ChR2-expressing and mCherry-expressing paternal voles with light on or light off. Paired sample *t*-test: ChR2 voles, t_off-on_ (4) = −2.792, *p* = 0.049 < 0.05; t_on-off_ (4) = 3.68, *p* = 0.021 < 0.05; mCherry voles, t_off-on_ (5) = −0.756, *p* = 0.484 > 0.05; t_on-off_ (5) = −1.446, *p* = 0.208 > 0.05. (**G**) The results of ChR2-expressing and mCherry-expressing infanticidal voles with light on or light off. Paired sample *t*-test: ChR2 voles, t_off-on_ (6) = −5.315, *p* = 0.002 < 0.01; t_on-off_ (6) = 5.362, *p* = 0.002 < 0.01; mCherry voles, t_off-on_ (5) = 0.0.045, *p* = 0.996 > 0.05; t_on-off_ (5) = −1.639, *p* = 0.162 > 0.05. The expression of the control virus and the effects of optogenetic activation of DR to MPOA 5-HTergic projection on other behaviors are shown in Appendix A. The injection sites and the diffusion of the virus in the ventral part of the DR are shown in Appendix A. The data are shown as mean ± SEM. ns, not significant; *, *p* < 0.05; **, *p* < 0.01.

## Data Availability

Data is contained within the article or Appendix A.

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
