# Peer review of "Involvement of Serotonergic Projections from the Dorsal Raphe to the Medial Preoptic Area in the Regulation of the Pup-Directed Paternal Response of Male Mandarin Voles"

_ijms, 2023, doi:10.3390/ijms241411605_

Round 1

Reviewer 1 Report

The study deals with a relevant topic with a sophisticated and well-applied methodology. The conclusions are supported by the data collected, and they deserve publication.

However, it is helpful to suggest minor additions at the introduction level, rationale for the study, and discussion of the results.

1) In the introduction, reference n°1 does not seem adequate to provide information on the care/aggression behaviors of young in the male of the human species, but rather to illustrate the effects of early child abuse. I suggest inserting specific human literature to relate to the phenomenon studied in the animal model.

2) In the rationale, at least on my part, the reason for choosing mandarine voles instead of the Californian mice with or without reproductive experience is not well appreciated, which would have produced the possibility of studying through which mechanisms the reproductive experience changes the phenotype from pup-attacking to pup-caring.

3) Finally, in the discussion, there is not the slightest hint of the translational relevance of these results. Is there any indication in humans of the involvement of serotonergic projections from the dorsal raphe to the medial preoptic area in regulating paternal/infanticide behavior?

Author Response

  • Comment 1 In the introduction, reference n°1 does not seem adequate to provide information on the care/aggression behaviors of young in the male of the human species, but rather to illustrate the effects of early child abuse. I suggest inserting specific human literature to relate to the phenomenon studied in the animal model.

Response: Thanks for your suggestion. We added a reference regarding the effects of father absence on mental health in human to embody the importance of fatherhood on the development of offspring. Thus, this sentence not only provides information on effects of aggression (child abuse) from father, but also provides information on effects of paternal care.

Culpin I, Heuvelman H, Rai D, Pearson RM, Joinson C, Heron J, Evans J, Kwong ASF. Father absence and trajectories of offspring mental health across adolescence and young adulthood: Findings from a UK-birth cohort. J Affect Disord. 2022 Oct 1;314:150-159. doi: 10.1016/j.jad.2022.07.016. Epub 2022 Jul 14. PMID: 35842065.

  • Comment 2: In the rationale, at least on my part, the reason for choosing mandarine voles instead of the Californian mice with or without reproductive experience is not well appreciated, which would have produced the possibility of studying through which mechanisms the reproductive experience changes the phenotype from pup-attacking to pup-caring.

Response: We appreciate your comments very much. Mandarin vole (Microtus mandarinus) is a monogamous rodent and males from this species display high levels of sociality and paternal care. This species is an ideal animal model for studying neural mechanism underlying paternal care. Californian mice are also monogamous species and distribute in America, but are not available for experiments in China. In our previous publications, changes in paternal responses and relevant neuroendocrine parameters induced by paternal experience have been investigated. It is found that levels of many neural chemicals such as central OT, TH and OTR, D1R, D2R mRNA expression have changed after becoming fathers (Wang et al., 2015,2018) These results have revealed the effects of fathering experiences. It is also found that virgin males showed different response to pups. Some attack or even kill the pups, while other take care of them. The differences in real time neural activity while males care or attack the pups, and neural circuit underlying different paternal responses directed by pups remains unclear. Although some studies observe father’s behaviors toward their own pups directly to investigate paternal care, most studies use virgin male to examine paternal behavior (Chauke et al., 2012). These between-individual variations provide an opportunity to reveal neural mechanisms underlying paternal care (Duclot et al., 2022). In the present study, paternal response of virgin male to pups were investigated. Thus, the findings from present study not only elucidate the mechanism underlying between-individual variation in pup-directed responsiveness of male, but also help us understand the mechanism underlying paternal care to some extent. It may also provide a new target for reduction of child maltreatment or abuse and increase of fathers’ engagement for therapeutic intervention of mental disease caused by these abnormal parental responses. This information has been added in the revised manuscript.

Duclot F, Liu Y, Saland SK, Wang Z, Kabbaj M. Transcriptomic analysis of paternal behaviors in prairie voles. BMC Genomics. 2022 Oct 1;23(1):679. doi: 10.1186/s12864-022-08912-y. PMID: 36183097; PMCID: PMC9526941.

Chauke, M.; de Jong, T.R.; Garland, T., Jr.; Saltzman, W. Paternal responsiveness is associated with, but not mediated by reduced neophobia in male California mice (Peromyscus californicus). Physiology & behavior 2012, 107, 65-75

Wang, B; Li, LF; He, ZX; Wang, LM; Zhang, SY; Qiao, H; Jia, R; Tai, FD. 2018. Effects of reproductive experience on paternal behavior, levels of testosterone, prolactin in serum and dendritic spines in medial prefrontal cortex of mandarin voles. INTEGRATIVE ZOOLOGY, 13: 711-722

Wang B, Li YN, Wu RY, Zhang SW, Tai FD.2015 Behavioral responses to pups in males with different reproductive experiences are associated with changes in central OT, TH and OTR, D1R, D2R mRNA expression in mandarin voles. Hormones and Behavior, 67, 73-82

  • Comment 3: Finally, in the discussion, there is not the slightest hint of the translational relevance of these results. Is there any indication in humans of the involvement of serotonergic projections from the dorsal raphe to the medial preoptic area in regulating paternal/infanticide behavior?

Response: The present study found that activation of 5-HTergic neural projections from DR to MPOA of virgin male could increase levels of paternal care and attenuate pup-attacking, while inhibition of these projection produced opposite effects revealing a possible mechanism underlying difference in pup-directed paternal responses. Thus, in the future, pharmacological manipulation of this pathway possibly can be used to reduce child abuse and increase fathers’ engagement to prevent or intervene the mental disease of offspring caused by abnormality in paternal responses. Thus. the present finding also provides a new target for prevention or treatment of some psychiatry diseases associated with abnormal parental responses to offspring such as child maltreatment or abuse, or lower levels of paternal care. The translational relevance of present finding has been added in the discussion. However, as far as we know, there is no any indication in human regarding involvement of this pathway in regulation of paternal/infanticide behavior. Because of limitation in noninvasive research techniques and ethics in human, findings on this aspect on human fall behind that in animals.

Reviewer 2 Report

This paper concludes that the serotonergic projections from vDR to MPOA regulate paternal response of virgin male mandarin voles. The authors showed that inhibition of serotonergic projection from vDR to MPOA decreased the paternal behaviors and shortened the latency to attack pups. On the other hand, activation of serotonergic neurons via optogenetics elongated licking duration and inhibited the infanticide. I consider that this study is generally conducted well. I have a major concern about sorting of paternal voles and infanticidal voles because changes in licking behavior were shown in paternal voles and attacking behavior changes were shown in infanticidal voles (Lines 74-76). The method of assorting animals should be described in the method in detail. These should be also explained in the abstract. Related to the above issue, something seems to be very wrong about the explanation of Figure 1F. Is this a comparison between licking behavior of Paternal vole and attacking behavior of infanticidal vole. How can different behaviors of different kind of voles (paternal and infanticidal voles) be compared? This result doesn’t make sense.

Line 194: Add “of” after inhibition.

Line 269-270: Weren’t there voles that ignore pups? How were they assorted?

I think English is Okay.

Author Response

Comment 1: I have a major concern about sorting of paternal voles and infanticidal voles because changes in licking behavior were shown in paternal voles and attacking behavior changes were shown in infanticidal voles (Lines 74-76). The method of assorting animals should be described in the method in detail. These should be also explained in the abstract. Related to the above issue, something seems to be very wrong about the explanation of Figure 1F. Is this a comparison between licking behavior of Paternal vole and attacking behavior of infanticidal vole. How can different behaviors of different kind of voles (paternal and infanticidal voles) be compared? This result doesn’t make sense.

Response: As reviewer suggested, the method of assorting animals has been described in the method in detail.This was also explained in the abstract.

We are sorry for the confusing from 1F.In this figure,  5-HT sensor 1.0 was utilized to monitor the release of 5-HT in the MPOA during different behaviors of male voles. The results show that paternal vole displayed significant increase of 5-HT while they licking pups, while infanticidal voles attacked pups, no changes in 5-HT release were observed.Changes in 5-HT release were also not found while paternal and infanticidal voles retrieved or sniffing pups. We concluded that 5-HT release increase specifically while paternal voles were licking pups.The F1 did not compare licking behavior of Paternal vole and attacking behavior of infanticidal vole, but compared 5-HT release while males licked or attacked pups.   

Before the behavior tests, the virgin male voles had thirty minutes to acclimatize the test room with the 80 Lux illumination. Then, an unrelated 7-day-old pup was then placed into the cage with a virgin male mandarin vole and all behavioral responses of the male voles were recorded for 10 min. Some males that licked,groomed or crouched on pups were considered as paternal voles while other males that attacked, bited pups were regarded as infanticidal voles. The paternal voles showed behaviors involving approaching, sniffing, retrieving, licking, grooming and crouching. The infanticidal voles showed behaviors including approaching, sniffing, retrieving, and attacking pups.

Comment 2: Line 194: Add “of” after inhibition.

Response: Done

Comment 3: Line 269-270: Weren’t there voles that ignore pups? How were they assorted?

Response :Thanks for your comment. The male mandarin voles seldomly ignore pups when they encounter pups first time. These males were not included in the further experiment and data analysis in the present study.

Round 2

Reviewer 2 Report

I consider that the authors have adequately addressed my concerns.

The quality of English is okay.